# MERGING FEED-FORWARD SUBLAYERS FOR COMPRESSED TRANSFORMERS

## ABSTRACT

With the rise and ubiquity of larger deep learning models, the need for high-quality compression techniques has been growing in order to deploy these models widely. The sheer parameter count of some models makes it difficult to fit them into the memory constraints of different hardware. In this work, we present a novel approach to model compression by *merging* similar parameter groups within a model, rather than pruning away less important parameters. Specifically, we propose a straightforward method for selecting, aligning, and merging separate feed-forward sublayers in Transformer models, and test our method on a language modeling task, image classification, and machine translation. With our method, we demonstrate performance comparable to the original models across our three diverse tasks while combining more than a third of model feed-forward sublayers. For instance, we can remove over 21% of total parameters from a Vision Transformer, while maintaining 99% of its original performance. Additionally, we observe that some feed-forward sublayers often exhibit regions of high similarity between their activations, which may help explain their surprising mergeability. [1]

## 1 INTRODUCTION

Recent advances in deep learning have been marked by a stark increase in model size in order to achieve state-of-the-art performance. With this trend towards increasing model parameter count to improve results, there is a need for more high-quality compression techniques that balance compression effectiveness with model capability. These techniques can help facilitate the use of these models across a variety of inference settings, depending on hardware availability.

Much of the prior work in model compression has built upon on distillation, quantization, and pruning techniques (Hinton et al., 2015; Fiesler et al., 1990; LeCun et al., 1989). Regarding pruning, prior work has introduced many approaches that identify regions of prunable parameters that can be removed from the model without drastically changing performance. These techniques can target individual neurons or general regions of a model—like attention heads, blocks from layers, or even entire layers. (Voita et al., 2019; Lagunas et al., 2021; Sajjad et al., 2023). However, while "unimportant" features have been important to identify for pruning techniques, we can also exploit the notion of "redundant" features as a compression target. There has been far less focus on compression methods that target potentially redundant features within a model.

When targeting redundant features for compression, we can turn to *merging* sets of similar parameters rather than pruning them. Relatedly, a line of recent work has explored merging parameters from two or more separate models in order to combine their functionalities into a single model (Goddard et al., 2024; Yang et al., 2024a). In our case, we can imagine extending parameter merging to merge *sublayers* within one model, rather than separate models, in order to achieve model compression.

To this end, we propose a novel compression method that aligns and merges several feed-forward sublayers within Transformer architectures (Vaswani et al., 2017). We target feed-forward sublayers in particular due to their large parameter count and easy mergeability. With a small amount of recovery fine-tuning, our models quickly regain competitive performance at reduced parameter counts. Through our testing, we find that these groups of feed-forward sublayers are notably compressible

---

[1]Toolkit and all experiments will be available at `code_link_forthcoming`

Figure 1: Overview of the feed-forward alignment and merging algorithm used to compress models in an example three layers of a Transformer. Multi-headed attention is abbreviated to MHA, feed-forward sublayers are depicted with $W^{\text{in}}$ and $W^{\text{out}}$ weights, and Add&Norm operations are depicted with $\oplus$, connected by arrows indicating residual connections.[2] Permutation transformation matrices are shown as $P_i$. Our method includes a permutation finding step, applying the transformations, merging transformed parameters, and finally tying the merged parameters. By merging and tying $k$ feed-forwards, we can reduce the model size by $k-1$ feed-forward sublayers.

via merging, giving rise to a simple and surprisingly effective framework, applicable to a wide variety of already-existing models.

We highlight the contributions of our work:

1. We propose a novel model compression method inspired by recent work in model merging. This approach is orthogonal to popular compression methods like quantization and pruning.

2. Across three different Transformer-based models, namely GPT-2, ViT, and a machine translation model, we show that merging over one-third of feed-forward sublayers and fine-tuning the resulting model can achieve performance comparable to the original models.

3. To explore the surprising effectiveness of merging, we compute similarity measures between feed-forward sublayers within the same model, and find regions with highly similar activations. These same patterns do not occur in attention sublayers.

## 2    RELATED WORK

In this section, we review work related to weight sharing for reduced parameter models, and prior work related to pruning and redundancy in models. We also summarize popular compression techniques in Table 1, and compare them to our merging-based compression approach.

### 2.1    WEIGHT SHARING FOR SMALLER MODELS

Prior work on weight sharing has largely focused on training models from scratch with specific sharing schemes. Sharing input and output embedding layers has been widely used to help cap total parameter count, but more importantly to provide important gradient sharing patterns for better generalization for many language tasks (Press & Wolf, 2017; Inan et al., 2017). In the case of non-embedding Transformer layers, prior work has explored numerous weight tying patterns for training new models (Dehghani et al., 2019; Reid et al., 2021; Takase & Kiyono, 2023). Liu et al. (2024) use heavy weight sharing at initialization between Transformer layers to achieve state-of-the-art sub-billion parameter language models. Pires et al. (2023) specifically tie feed-forward sublayers at initialization and train

---

[2]This diagram shows a Post-LN Transformer, but our method easily applies to Pre-LN Transformers as well.

machine translation models that can outperform standard Transformer architectures, when training at large enough tied feed-forward widths. In this work, we instead start from a pre-trained model, and then use weight sharing as a tool to reduce the overall parameter count.

## 2.2 Pruning and redundancy

Prior work has explored different aspects of redundancy patterns between adjacent Transformer components, and suggested several techniques to reduce or exploit this phenomenon. Dalvi et al. (2020) use CKA to track layer redundancy in BERT and XLNet and correlation clustering to find redundant sets of neurons. Using the discovered clusters, they remove redundant neurons for fewer total parameters. Men et al. (2024); Gromov et al. (2024) show that by removing entire Transformer layers in decoder-only models, very deep language models can achieve inference speedups without sacrificing major performance. Li et al. (2024) propose a compression method applicable to sparsely-activated mixture-of-expert models (SMoEs) that similarly draws from model merging work to compress experts in large SMoE models. Our method extends a similar approach to a much wider set of compressible models.

Table 1: A summary and comparison of different compression methods, including merging.

|  | **Motivation** | **Training Required** | **Run Time Savings** |
|---|---|---|---|
| Quantization | reduce precision | No | No[3] |
| Pruning | remove unimportant parameters | generally fine-tuning | Depends |
| Distillation | train smaller student from teacher | Yes | Yes |
| Merging | combine redundant parameters | fine-tuning | No |

## 3 Merging Feed-Forward Sublayers

In this section, we discuss choosing feed-forward sublayers as our merging target, review necessary background for permutation-based neuron alignment, and then describe our compression method.

### 3.1 Feed-forward sublayers as a merging target

We focus our interest on Transformer feed-forward (FF) sublayers for several reasons. The first, and most simple, is that these sublayers generally constitute around two-thirds of non-embedding parameters Transformer encoder or decoder models. Compressing these parameters can constitute substantial overall savings in a model. Secondly, as we consider merging these parameters, we note that the parameterization of FF sublayers is far simpler than the other major sub-block of a Transformer layer, namely multi-headed attention (MHA). This structural simplicity makes it a good candidate for merging-based approaches for compression.

Beyond practical considerations, prior work has also established several properties of Transformer FF sublayers that make them good candidates for compression via merging. Prior work on FF sublayers has shown that they can be very sparsely activated (Li et al., 2023), where non-zero percentages of FF activations can be as low as 3-5%. Additionally, another work has demonstrated evidence that combining LayerNorm and feed-forwards, in both Post- and Pre-LN architectures, results in some weakening effects of the contextualization effects of FF sublayers (Kobayashi et al., 2024). The authors allude to redundancy in the Transformer's processing due to this interaction. Finally, Pires et al. (2023) train Transformer-based translation models with only one feed-forward sublayer in the encoder, tied across each layer. Their models, when trained with extended FF widths, can outperform base transformers at the same parameter budget.

---

[3] Quantization can improve batch throughput during inference, which can result in run time savings, but it generally does not improve inference speed at a constant batch size.

### 3.2 BACKGROUND ON PERMUTATION-BASED NEURON ALIGNMENT

We propose a merging technique that combines several similar sublayers into a single parameter set. Our merging technique is inspired by prior work in permutation symmetries of neurons (Li et al., 2015). This type of prior work has been widely used in studying convergent learning between models, as well as performing model merging between two or more separate models (Tatro et al., 2020; Entezari et al., 2022; Ainsworth et al., 2023).

Permutation-based neuron alignment techniques seek to find a superior ordering of neurons in one layer in order to more closely match the ordering of neurons from another layer. Given two layers with neurons we wish to align, we compute a forward pass through both of these layers using relevant data in order to collect features. The layers are generally corresponding parameters from two different models. We collect these sets of activations, $X_\alpha, X_\beta \in \mathbb{R}^{n \times d}$, from the output of the two parameter sets, where $n$ is the number of tokens or patches processed in the forward pass, and $d$ is the feature dimension.

To determine corresponding features from these activation sets, we compute cross-correlation $C$, in line with prior work (Li et al., 2015). $\mu$ represents mean vectors, and $\sigma$ standard deviation vectors.

$$C = \text{corr}(X_\alpha, X_\beta) = \frac{\mathbb{E}\left[(X_\alpha - \mu(X_\alpha))^T (X_\beta - \mu(X_\beta))\right]}{\sigma(X_\alpha)\sigma(X_\beta)} \tag{1}$$

The resulting matrix $C \in \mathbb{R}^{d \times d}$ reflects how each feature $j$ in $X_\alpha$ correlates with each feature $k$ in $X_\beta$. Now, to find the mapping of features from $X_\alpha$ to $X_\beta$ that maximize overall correlation, we solve the following optimization problem, where $\Pi_d$ is the space of all permutations of length $d$ (Li et al., 2015; Tatro et al., 2020):

$$\pi^* = \max_{\pi \in \Pi_d} \sum_{j=1}^{d} C(j, \pi(j)) \tag{2}$$

This problem is a case of the Linear Assignment Problem (LAP), and we solve for $\pi^*$ using the Jonker-Volgenant algorithm implementation provided by `scipy` (Crouse, 2016).

### 3.3 COMBINING FEED-FORWARD SUBLAYERS

Now, with the appropriate background, we describe our compression method. For our method, we first assume that we have some predetermined number of feed-forward sublayers $k$ that we want to merge. This number can be inferred if given a goal overall parameter reduction ratio, or set otherwise. In summary, our compression method aligns the ordering of the neurons between the two feed-forward sublayers in order to merge them.

Given a window of $k$ adjacent feed-forward sublayers, we compute a forward pass using a subset of data in order to compute features for each feed-forward hidden state. In other words, for Transformer FF sublayer $x^{\text{out}} = W^{\text{out}}\phi(W^{\text{in}}x^{\text{in}} + b^{\text{in}}) + b^{\text{out}}$, we obtain features just before the $\phi$ activation. We consider only the neurons just *after* $W^{\text{in}}$ because prior work has shown that to reorder the input to $W^{\text{in}}$ and output of $W^{\text{out}}$ requires permuting many additional weights due to the residual connections in order to maintain functional equivalence (Verma & Elbayad, 2024). For each of the $k$ feed-forward sublayers, we collect features $X_i \in \mathbb{R}^{n \times d}$ $i \in [0, k-1]$, where $n$ is the number of tokens or patches processed, and $d$ is the feed-forward dimension.[4]

We designate the first feed-forward sublayer of the set to be our "anchor", and we compute the permutation finding algorithm on each pair of feed-forwards where the first item is always the anchor. In other words, for each sublayer $i \in [1, k-1]$, we have inputs $X_0$ and $X_i$, and find $\pi_i$ using the permutation finding algorithm from Section 3.2.

After converting function $\pi_i$ to its corresponding permutation matrix $P_i$, we can apply them to each of the corresponding $k-1$ feed-forward sublayers. We then average the transformed weight matrices, and replace each of the $k$ feed-forwards with their average, as in Equation sets 3 and 4. Finally, we tie these weights so that in memory they appear as just one sublayer, effectively removing the parameters

---

[4]The layer indices reflect local index within the set of $k$ versus global layer index.

from $k - 1$ feed-forward sublayers.

$$W^{\text{in}*} = \frac{1}{k} \left( W_0^{\text{in}} + \sum_{i=1}^{k-1} P_i W_i^{\text{in}} \right) \qquad\qquad b^{\text{in}*} = \frac{1}{k} \left( b_0^{\text{in}} + \sum_{i=1}^{k-1} P_i b_i^{\text{in}} \right) \qquad (3)$$

$$W^{\text{out}*} = \frac{1}{k} \left( W_0^{\text{out}} + \sum_{i=1}^{k-1} W_i^{\text{out}} P_i^T \right) \qquad\qquad b^{\text{out}} = \frac{1}{k} \left( \sum_{i=0}^{k-1} b_i^{\text{out}} \right) \qquad (4)$$

## 3.4 Selecting sublayers to merge

In selecting the $k$ adjacent feed-forward sublayers to merge, we take a sliding window approach. For all starting layer indices from 0 to $(N_{\text{layers}} - 1) - k$, we apply the method outlined in Section 3.3, and evaluate the resulting compressed model on a validation set.

In reality, although we propose to test each potential window, the cost of computing permutations and parameter arithmetic is low. The largest cost each iteration is computing features and testing candidates. However, we only compute features *once* despite testing $N_{\text{layers}} - k$ models, because one forward pass through the exemplar data is sufficient for creating all necessary correlation matrices. We test these potential candidates and choose the one with best starting evaluation score. We note that there may be other possible selection heuristics in this setting.

Finally, we follow our merging procedure with recovery fine-tuning to quickly heal performance on the downstream task. We include an algorithm for our selection method in Algorithm 1.

---

**Algorithm 1** Feed-Forward Sublayer Merge

---

**Input:** Model parameters $\theta_{\text{input}}$, collected features $\{X_i\}_{i=0}^{N_{\text{layers}}-1}$, batched fine-tuning data $D_{ft}$
**Input constants:** $k$, $N_{\text{layers}}$, MAXUPDATES
**Initialize:** $\theta_{\text{selected}}$, BESTSCORE $\leftarrow 0$           // Assuming a maximized score
**for** $i = 0$ **to** $(N_{\text{layers}} - 1) - k$ **do**
    $\theta_{\text{merged}} \leftarrow$ COMPRESS$(\theta_{\text{input}}, \{X_i\}_{i=0}^{N_{\text{layers}}-1}, k)$
    **if** EVAL$(\theta_{\text{merged}}) >$ BESTSCORE **then**
        $\theta_{\text{selected}} \leftarrow \theta_{\text{merged}}$
    **end if**
**end for**
**for** $i = 0$ **to** MAXUPDATES **do**
    $\theta_{\text{selected}} \leftarrow$ UPDATE$(\theta_{\text{selected}}, D_{ft}(i))$           // Fine-tuning step
**end for**
**Output:** $\theta_{\text{selected}}$

---

## 4 Experimental Setup

For testing the extensibility of our method, we test our compression method on several different Transformer-based models. Specifically, we use GPT-2 (Radford et al., 2019), the Vision Transformer (ViT) (Dosovitskiy et al., 2020), and a Transformer-based machine translation model from OPUS-MT (Tiedemann & Thottingal, 2020). We use this variety of models in order to cover a diversity of model types (decoder-only, encoder, encoder-decoder) and different modalities.

For each setting, we list the exact model used, the data used to compute example features for correlations, and finally the data used for recovery fine-tuning and evaluation. Additional fine-tuning hyperparameters are included in Appendix B.

## 4.1 Language modeling

For our experiments, we use the large release of GPT-2, which has 36 layers and a feed-forward dimension of 5120. For computing feed-forward features, we use 10k tokens from the validation set of the Wikitext103 dataset (Merity et al., 2017). Finally, we use the train and test sets from the same Wikitext103 for fine-tuning and evaluation, respectively.

Unlike the other two tasks, the pre-training data for GPT-2 is not publicly available, so we use Wikitext103 training data for fine-tuning. Due to this discrepancy, our uncompressed GPT-2 baseline is also fine-tuned on Wikitext103 train in order to make a fair comparison. Because we have access to the training data for the machine translation and ViT models, we do not provide a fine-tuned baseline for those as the data we use already appears in their original training data.

We fine-tune our GPT-2 models for up to 100k steps with a batch size of 2. We pack batches to the context length of 1024 after tokenization with the GPT-2 tokenizer. We select the best model based on validation perplexity and report average test perplexity with a sliding window of 512 tokens.

### 4.2 IMAGE CLASSIFICATION WITH VIT

We use a vision transformer (ViT) for our image classification experiments, with resolution of 224x224, and patch size of 16x16.. ViT is a 12-layer Transformer Encoder architecture that is pre-trained on ImageNet-21k, and subsequently fine-tuned on ImageNet-1k. ImageNet-1k is a classification task where images belong to one of 1000 categories (Russakovsky et al., 2015). For computing feed-forward features, we use 10k patches from the ImageNet-1k validation set. Evaluation results are computed on original validation labels.

We fine-tune our ViT models on ImageNet-1k train for up to 50k steps with a batch size of 128, and report accuracy scores.

### 4.3 MACHINE TRANSLATION

For our experiments on machine translation, we use a Chinese-English model from the OPUS-MT release (Tiedemann & Thottingal, 2020). It is a 12-layer encoder-decoder Transformer with cross-attention. For computing feed-forward features, we use 10k tokens from the Tatoeba validation set, counted on the source side (Tiedemann, 2020). For fine-tuning, we use the original training data released by the Tatoeba translation challenge, sourced from OPUS (Tiedemann, 2012). We apply our method to both the encoder and decoder separately, constituting two anchors. However, we search windows in sync, meaning that the same window from the encoder and decoder are merged.

We fine-tune our translation models for up to 100k steps with a batch size of 64. We use `sacrebleu` to compute BLEU scores for evaluation (Papineni et al., 2002; Post, 2018).

### 4.4 LAYER PRUNING BASELINE

Recent work on structured pruning of Transformers has been marked by a large number of methods presenting ways to remove adjacent layers from models and then optionally fine-tune the compressed model (Men et al., 2024; Gromov et al., 2024; Yang et al., 2024b). We focus on structured pruning baselines as many unstructured pruning baselines do not necessarily lead to smaller memory requirements unless they achieve 1) high sparsity ratios and 2) use specialized sparse libraries to store sparse model weights. Our method, on the other hand, is realized memory-wise as a smaller model due to shared weights only being stored once.

Many layer-pruning methods rely on a similarity technique to choose a subset of adjacent and similar layers to prune. However, in our baseline, we forgo any specific similarity techniques to choose a subset, and instead choose the best possible subset, much like our own technique, via a sliding window. After selecting the best window after evaluation, we then fine-tune the model with the same specifications as our method. In all, this encapsulates a strong, structured pruning baseline that generalizes many layer-pruning based techniques.

## 5 RESULTS

### 5.1 MERGING FEED-FORWARD SUBLAYERS ACROSS COMPRESSION RATIOS

We evaluate our compression method on image classification using ViT, language modeling using GPT-2, and machine translation using an OPUS-MT zh-en model, and report our results in Figure 2. We report results at 1/3, 1/2 and $(n-1)/n$ feed-forward sublayers removed, in order to test our

method at different overall compression ratios.[5] We also report results from our compression method without the permutation step, as seen as "Vanilla" in the figure.

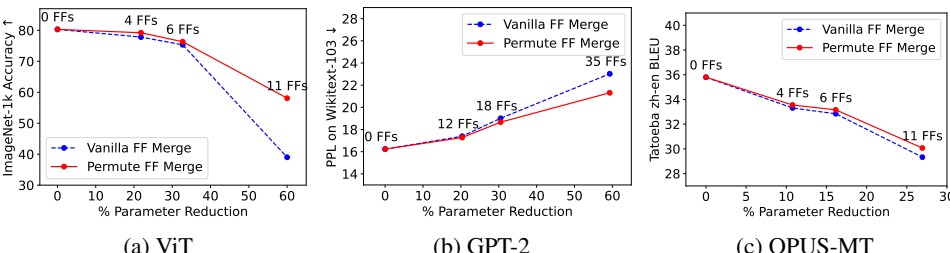

(a) ViT  (b) GPT-2  (c) OPUS-MT

Figure 2: Results across all three tasks depicting compression versus performance results. We include results from our main method, labeled as Permute FF Merge, as well as our method without permutation alignment, depicted as Vanilla FF Merge. We note that our method retains almost complete performance at one-third of feed-forward sublayers removed, across all tasks, and continues to retain high performance at one-half of FF sublayers removed.

From our results, we see that even up to 1/2 of feed-forward sublayer parameters removed, which is over 30% in parameter reduction for ViT and GPT-2,[6] our method can retain high performance, similar to the base model. At 1/3 of feed-forward sublayers removed, performance is almost identical, resulting in only a 1% accuracy drop in ViT, 1 PPL increase in GPT-2, and 2 BLEU drop in the translation model. Full numerical results can be found in Appendix A. We note that in this sub-billion parameter regime, prior work has shown that smaller models are more difficult targets of compression methods (Ashkboos et al., 2024), as well as dense models versus models with natural sparsity patterns (i.e. Mixture-of-Expert models).

Our findings also hold across all three of our tasks tested, suggesting that our method generalizes to different types of Transformer-based models. Additionally, we can notice that permutation-based compression is consistently better compared to vanilla averaging compression, demonstrating the effectiveness of aligning features within feed-forward sublayers before merging them. This effectiveness is more pronounced at larger numbers of feed-forward sublayers removed. In summary, our results show that 1) post-training weight sharing is a simple and effective compression method and 2) permutation-based alignment of these shared weights can improve final compression performance.

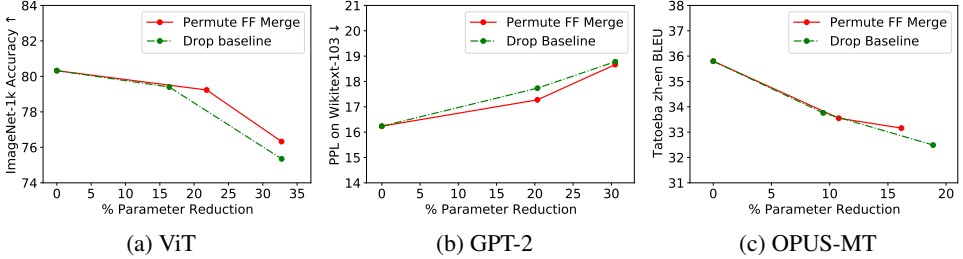

(a) ViT  (b) GPT-2  (c) OPUS-MT

Figure 3: Results across all three tasks depicting compression versus performance for our method and a strong layer-dropping baseline method. We perform layer dropping for 1/6 and 1/3 of layers dropped, and fine-tune the best pre-tuned set of dropped layers for all sliding windows. Across the parameter reduction range shown, our merging-based compression method outperforms or matches layer-dropping across the three tasks.

In Figure 3, we compare our method at 1/3 and 1/2 FFs removed to our layer-pruning baseline. We drop layers to attempt to match the reduction ratios of our own methods, constituting 1/6 and

---

[5]We note that the OPUS-MT compression ratios are different due to the additional presence of cross-attention in enc-dec architectures.

[6]We include embedding parameters in all % parameter reduction and compression ratio calculations

1/3 layers dropped for all three models. However, since we cannot match exact ratios, we plot the exact parameter reduction ratios and performance, and compare. As seen in the figure, our method consistently matches or outperforms the layer-dropping method. This comparison confirms that merging is a competitive alternative to strong pruning-based methods for model compression.

## 5.2 CHOICE OF MERGED SUBLAYERS

In our merging algorithm, we choose which layers to merge by computing performance over a sliding window of $k$ indices. In doing this, we observe the performance for each set of adjacent feed-forward groups. For each of our model/task pairs, we plot the performance of the merging algorithm on 1/3 of feed-forward sublayers before tuning across all groupings, to observe the differences across these groups. Results are shown in Figure 4. Before tuning, it appears that the choice of layers seems to be important, resulting in different performance.

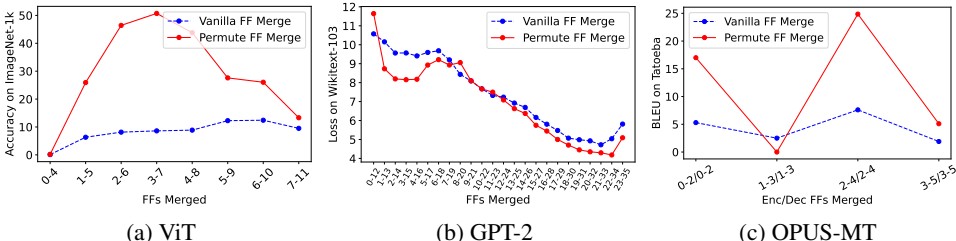

| (a) ViT | (b) GPT-2 | (c) OPUS-MT |

Figure 4: Performance curves over different ranges of merged feed-forward sublayers representing 1/3 merged. Across all three tasks, there are clear ranges of merged feed-forward sublayers that retain more performance when merged.[7]

However, these differences reduce once recovery fine-tuning is performed. To see this, we randomly select 3 sets of $k$ consecutive layers for each of our tasks, and apply recovery fine-tuning to these compressed models. In Table 2, we observe that all models achieve similar performance after fine-tuning. Nevertheless, the choice of layers might be important if non-adjacent merges are allowed; this is potential future work.

Table 2: Results comparing our compression method with 1/3 of feed-forward sublayers removed, but with different sublayer groups. We include three random consecutive selections of sublayers, excluding the original selection.

| Model | Metric | Best pre-tune | Random 1 | Random 2 | Random 3 |
|---|---|---|---|---|---|
| ViT | Accuracy(%) ↑ | 79.2 | 79.5 | 78.5 | 78.9 |
| GPT-2 | PPL ↓ | 17.3 | 18.3 | 17.1 | 17.3 |
| OPUS-MT | BLEU ↑ | 33.6 | 33.9 | 33.8 | 33.1 |

## 5.3 CHOICE OF ANCHOR LAYER

In addition to analyzing the subset of layers to merge, we also wish to understand the sensitivity of our merging compression method to the choice of anchor layer for our alignment step. In section 3.3, we choose the first feed-forward sublayer in the sequence to serve as the reference, and compute permutations aligning the following sublayers to this reference. Here, we additionally consider using either the *last* of the sequence, or the *middle* of the sequence, and report results in our 1/3 feed-forward merge setting in Table 3.

Similarly to the choice of layers, our merging approach is robust to the choice of reference or anchor layer. This indicates that our method is not overly sensitive to the choice of which sublayer to align other sublayers to, enhancing the reliability of our permutation-based alignment method to find corresponding features for a useful merge.

---

[7]We display loss on Wikitext-103 for visibility.

Table 3: Results comparing our compression method with 1/3 of feed-forward sublayers removed, but with different anchor locations.

| Model | Metric | Anchor First | Anchor Middle | Anchor Last |
|---|---|---|---|---|
| ViT | Accuracy(%) ↑ | 79.2 | 79.5 | 79.0 |
| GPT-2 | PPL ↓ | 17.3 | 17.4 | 17.4 |
| OPUS-MT | BLEU ↑ | 33.6 | 33.4 | 33.5 |

## 5.4 ADDITIONAL COMPRESSION VIA QUANTIZATION

We are interested in seeing if our method works well in combination with other compression methods. For example, quantization is an extremely effective method to reduce the memory footprint of a model by reducing the numerical precision of the parameters. While our compression method focuses on identifying redundancies to reduce the overall parameter count via parameter sharing, quantization can help reduce the overall storage needed for a model, and still proves an extremely effective compression technique. Therefore, we wish to ensure that our method performs orthogonally to state-of-the-art quantization, so that both methods may be used for additional storage savings.

We experiment with the LLM.int8() quantization method due to its effectiveness and widespread adoption (Dettmers et al., 2022). In brief, this method extends absmax quantization, but retains 16-bit precision for outlier values. We quantize our models after removing 1/3 of feed-forward sublayers, and report scores in Table 4.

Table 4: Compression results across three tasks, before and after additional compression via quantization. In this case, compression is measured in terms of total model storage complexity (disk space) instead of parameter count.

| Model | Metric | Our Method | | +LLM.int8() | |
|---|---|---|---|---|---|
| | | Compression | Performance | Compression | Performance |
| ViT | Accuracy(%) ↑ | 78% | 79.2 | 20% | 79.2 |
| GPT-2 | PPL ↓ | 80% | 17.3 | 22% | 17.3 |
| OPUS-MT | BLEU ↑ | 89% | 33.6 | 51% | 33.6 |

Combining our method with quantization provides an even smaller compression ratio, while retaining high performance. Coupling quantization with additional compression, like our method, helps to realize compression ratios like 20% when considering total model storage complexity.

## 5.5 SIMILARITY TRENDS ACROSS FEED-FORWARD SUBLAYERS

So far, we have shown that simply aligning, merging, and tying adjacent feed-forward sublayers is a simple, yet effective technique for compressing Transformer models. Because of this, we look further into the similarities between representations computed from different feed-forward sublayers. We are interested in if these sublayers exhibit signs of redundancy, as eluded to in previous work (Pires et al., 2023; Kobayashi et al., 2024).

To this end, we compare outputs between feed-forward sublayers within the same models. Across our three tasks, we use 10k tokens or patches, depending on the model, from task validation sets to compute a set of output states from all feed-forward sublayers. Then, we use Centered Kernel Alignment (CKA) to compute their similarity. CKA is a state-of-the-art method for comparing the similarity between neural network activations (Kornblith et al., 2019). We plot CKA similarity values for all pairwise interactions between FF sublayers in all three of our model types, shown in Figure 5.

We notice that across all three model/task pairs, clear regions of high similarity can be observed. This means that the outputs of these feed-forward sublayers are highly similar, despite being interleaved with multi-headed attention sublayers. We note that similar behavior is not seen in attention sublayers, as seen in Appendix C. While prior work has shown similarities between the outputs of adjacent *full* Transformer layers, this similarity can be explained in part to the residual computations that add

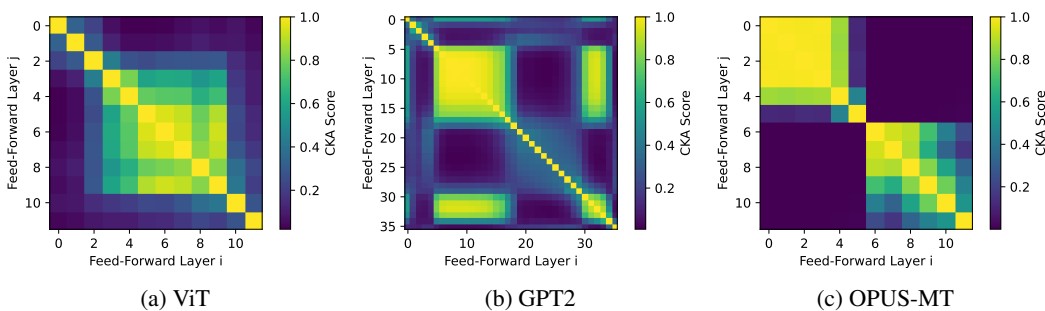

(a) ViT             (b) GPT2             (c) OPUS-MT

Figure 5: CKA plots of feed-forward sublayer hidden states across three different models. In all three settings, we see clear regions of high similarity between different FF layers. We do not compare between encoder and decoder feed-forward sublayers in the Translation model due the differences in token inputs.

the prior sublayer output to the current sublayer output (Kornblith et al., 2019; Dalvi et al., 2020). However, in comparing feed-forward outputs, we isolate this signal from the stream of residual computations, before the output is added back to the input, making the observed similarity more surprising due to the greater independence between these computations.

## 6 CONCLUSION

In this work, we propose a novel compression method that applies to Transformer-based models via merging and tying adjacent sets of feed-forward blocks. Our method serves as an alternative to existing compression approaches, and opens possibilities of future methods that examine the use of parameter merging and weight tying as a post-training compression technique in deep learning. We demonstrate our method's extensibility by applying it to several types of Transformer-based models, namely GPT-2, ViT, and an OPUS-MT translation model. Across these diverse tasks, we show that our method can maintain almost full performance while removing 1/3 of feed-forward sublayers, and maintains high performance even after removing 1/2 of all feed-forward sublayers. Finally, we explore the differences in representation similarity between feed-forward and attention sublayers, and find regions of high similarity between feed-forward sublayers (despite being separated by attention sublayers), which may be related to their surprising mergeability found in our experimentation.

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

# A  FULL RESULTS AT VARYING COMPRESSION RATIOS

Table 5: Full numerical results on compression results at 1/3 FF sublayers removed, 1/2 FF sublayers removed, and $(n-1)/n$ FF sublayers removed. Original, uncompressed models are included in the first row of results for each model, indicated by 0 FFs removed and no merged indices.

| Model | Metric | Merged Indices | FFs Removed | Vanilla | Permute |
|-------|--------|----------------|-------------|---------|---------|
| ViT | Accuracy (%) ↑ | – | 0/12 | 80.3 | 80.3 |
|  |  | 3-7 | 4/12 | 77.8 | 79.2 |
|  |  | 4-10 | 6/12 | 75.3 | 76.3 |
|  |  | 0-11 | 11/12 | 39.0 | 58.1 |
| GPT-2 | PPL ↓ | – | 0/36 | 16.16 | 16.16 |
|  |  | 22-34 | 12/36 | 17.39 | 17.27 |
|  |  | 16-34 | 18/36 | 19.01 | 18.66 |
|  |  | 0-35 | 35/36 | 23.02 | 21.31 |
| OPUS-MT | BLEU ↑ | – | 0/12 | 35.8 | 35.8 |
|  |  | 2-4/2-4 | 4/12 | 33.3 | 33.6 |
|  |  | 0-3/0-3 | 6/12 | 32.8 | 33.2 |
|  |  | 0-5/0-5 | 11/12 | 29.3 | 30.1 |

# B  FINE-TUNING DETAILS

## B.1  GPT-2

Table 6: Hyperparameters used for GPT-2 fine-tuning

| Hyperparameter | Value |
|----------------|-------|
| Start LR | 5e-5 |
| LR Schedule | inv_sqrt |
| fp16 | True |
| batch size | 2 |
| n_steps | 100K |

## B.2  VIT

Table 7: Hyperparameters used for ViT fine-tuning

| Hyperparameter | Value |
|----------------|-------|
| Start LR | 5e-5 |
| LR Schedule | lin_decay with min |
| decay_steps | 20K |
| Min LR | 1e-6 |
| fp16 | True |
| batch size | 128 |
| n_steps | 50K |

## B.3  MACHINE TRANSLATION

We select our best model using validation BLEU, computed on a 2000 instance subset of the full Tatoeba validation set.

Table 8: Hyperparameters used for OPUS-MT fine-tuning

| Hyperparameter | Value |
| --- | --- |
| Start LR | 5e-5 |
| LR Schedule | inv_sqrt |
| fp16 | True |
| bsz | 64 |
| n_steps | 100K |

## C ATTENTION LAYER SIMILARITY

We compute CKA similarity between all attention sublayer pairs, using the same 10k tokens or patches from our CKA results on FF sublayers. The features are from the output of the linear layer just after the dot-product attention computation.

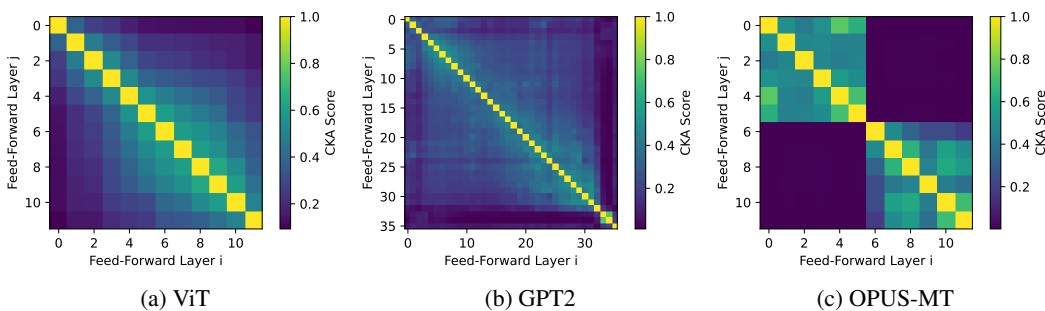

(a) ViT          (b) GPT2          (c) OPUS-MT

Figure 6: CKA plots of multi-headed self-attention sublayer activations across three different trained models. Attention activations are largely dissimilar from each other across model types. We do not compare between encoder and decoder attention sublayers in the translation model due the differences in token inputs.

