# OpenReview forum: "Merging Feed-Forward Sublayers for Compressed Transformers"
_ICLR.cc/2025/Conference — ICLR 2025 Conference Withdrawn Submission_

### Official Review · Reviewer_65Xr · 2024-10-19

**Soundness:** 2
**Presentation:** 1
**Contribution:** 2
**Rating:** 3
**Confidence:** 5

**Summary:**

This paper proposes a method to compress Transformer models by merging similar feed-forward network (FFN) layers. The authors align and average the weights of these layers to reduce parameter count by up to 30%, without significantly impacting model performance.

**Strengths:**

- The paper presents a clear and straightforward idea. The authors propose reusing the Feed-Forward Network (FFN) layers in Transformer models, which makes the paper relatively easy to understand. The main novelty comes from averaging FFN weights after applying permutation to align them across layers.
- The proposed method demonstrates that model compression can be achieved by reducing the number of stored parameters through merging FFN layers. This can be useful for reducing memory usage in models deployed on hardware with storage limitations.

**Weaknesses:**

- **Limited Practical Use:** The approach only reduces the number of stored parameters without reducing computational cost (no FLOP reduction). This is a significant limitation because many existing compression techniques like pruning aim to reduce both memory and computation, enabling models to run on resource-constrained devices with lower latency. The authors' method, while helpful in reducing memory, doesn't address this more practical need, limiting its applicability.
- **Lack of Comprehensive Baselines:** The experimental evaluation is insufficient, as it misses several important baselines:
    - **Naive Baseline:** A simple baseline, such as reusing the FFN layers and fine-tuning the shared parameters from a random initialization, would be helpful to assess the effectiveness of the proposed approach.
    - **Pruning Methods**: Methods like magnitude pruning or zero-activation pruning should be included to compare accuracy under the same parameter reduction.
    - **Methods Targeting FFN Optimization**: Other methods specifically focused on reducing computation and memory usage of FFN layers, such as MoEfication​ [1] and Go Wider Instead of Deeper​ [2], should be used as baselines as well.
- **Writing Quality:** The paper contains several writing issues. For example, line 173 states "from the output of the parameter two parameter sets," which is confusing and unclear. Similarly, line 156 misuses "elude," where "allude" would be more appropriate.

[1] Zhang Z, Lin Y, Liu Z, et al. Moefication: Transformer feed-forward layers are mixtures of experts. arXiv 2021.
[2] Xue F, Shi Z, Wei F, et al. Go wider instead of deeper. AAAI 2022.

**Questions:**

- In line 89, the authors claim that "These same patterns do not in counterpart attention sublayers." However, there are works like [3] and [4] that show how attention weights can also be reused across layers. Could the authors clarify this discrepancy?
- Equation (4): Why does the equation for the bias term $b_i^{out}$ not include the permutation matrix $P_i$?
- The "Vanilla" version in Figure 2 is described as "without the permutation step." Does this mean that the merged FFN weights are simply averaged without any alignment?
- What happens if fine-tuning is not applied after merging the FFN layers? How much does fine-tuning contribute to the performance recovery?
- Why do the authors use a sliding window to select consecutive layers for merging? Wouldn't a strategy based on similarity metrics across non-consecutive layers be more effective?

[3] Xiao T, Li Y, Zhu J, et al. Sharing attention weights for fast transformer. arXiv 2019.
[4] Bhojanapalli S, Chakrabarti A, Veit A, et al. Leveraging redundancy in attention with reuse transformers. arXiv 2021.

---

> ### Author Response · Authors · 2024-11-26
> **Response to  Reviewer 65Xr**
>
> Thanks for your time and thoughtful review of our paper. We address the stated weaknesses and questions below:
>
> **Weakness 1: limited practical use**
>
> Compressing models without necessarily reducing speed is still a major practical contribution. While we agree that structured pruning methods do frequently result in some speed increase, many methods exist to train or compress models to achieve small sizes without focusing on run-time as well. For example, small language models with built-in weight sharing, some unstructured pruning methods and some quantization methods share this property. For example, LLM.int8 quantization is actually slower than its unquantized counterpart for GPT-Large.
>
> **Weakness 2: More baselines**:
>
> Thanks for your comment regarding comparisons and discussions. We agree that even if our method takes a very different approach to compression, it should be compared to alternatives. Since we propose a general compression method not specific to ViT, we choose a general pruning method for comparison. Many recent papers on structured pruning of Transformers (which results in compression unlike many unstructured pruning methods) have centered around choosing layers for dropping and then fine-tuning the resulting model [1,2,3]. We use a strong baseline of 1) picking the best layers to drop after evaluation (which generalizes and strengthens many of these layer-dropping papers) and 2) fine-tuning the same as our method, for all parameters. We choose the number of layers dropped to cover a similar range as ⅓ and ½ FFs removed. We present the results in the revised PDF (Section 4.4 details, Section 5.1 results) and link them here: https://anonymous.4open.science/r/temp-C34A/README.md . We outperform or match the baseline consistently judging by the curves across different parameter reduction ratios. Although we cannot achieve exact comparisons across specific parameter reduction ratios due to the block wise reduction nature of both methods, our method trends better in general, as seen in the figure.
>
> **Weakness 3: Writing quality**:
>
> Thanks for pointing out these 2 typos. We have corrected them in the updated PDF and rechecked thoroughly for others. However, we would like to point out that reviewer zq1y has stated the paper is well-written. We hope that this is just a minor issue at the typo level rather than something more systematic. We are happy to receive further feedback if it happens to be the latter.
>
> **Question 1: Other attention conclusions**:
>
> The first cited work re-trains only translation models from scratch with shared attention weights. Additionally, their similarity analysis is on *weight distributions* rather than the similarity between attention activations. Training the model from scratch with this inductive bias is very different than our comparison between attention layers, and seemingly easier to reuse attention states. The second cited work compares *attention score matrices* between layers whereas we compare attention output activations just after the linear projection following multi-headed attention. The output incorporates also the value vectors, as well as the output projection, leading to a very different output than the attention score matrix. In summary, direct similarity evaluations are applying on different objects in all 3 papers, and cited paper 1 is training models from scratch with this sharing paradigm.
>
> **Question 2: Why no P matrix in $b^{\text{out}}$ equation**:
>
> The $P_i^T$ matrix applies to the input dimension of the $W_i^{\text{out}}$ matrix, whereas the bias term is added to the output of the $W_i^{text{out}}$ projection. For FF sublayer $x_\text{out} = W_{\text{out}} \sigma(W_\text{in} + b_\text{in}) + b_{\text{out}}$, applying permutations yields $x_\text{out} = W_{\text{out}}P^T \sigma(P(W_\text{in} + b_\text{in})) + b_{\text{out}}$, showing this more clearly.
>
> **Question 3: vanilla baseline clarification**:
>
> Yes, this is correct. We include this baseline to observe the effect of the permutation alignment.
>
> **Question 4: Pre-tuning results**:
>
> These results are in section 5.2. Fine-tuning clearly contributes to the performance recovery, but we limit the amount of fine-tuning as described in Section 4. Through our results and analysis, we show that the permutation alignment and weight sharing provides a solid starting point for limited downstream fine-tuning.
>
> **Question 5: Sliding window**:
>
> We use the sliding window strategy to exhaust all sets of k adjacent feed-forward layers. We require adjacency for 1) combinatorial ease (ex, 36 GPT-2 layers choose 12 is > 1B) as well as some evidence of similarity that aligns with adjacency from prior work [1, 2] as well as our own (i.e. Figure 4).
>
> [1] Pires et al. "One wide feedforward is all you need." arXiv preprint arXiv:2309.01826 (2023).
>
> [2] Kornblith, Simon, et al. "Similarity of neural network representations revisited." International conference on machine learning (ICML), 2019.

---

### Official Review · Reviewer_SKiK · 2024-10-31

**Soundness:** 2
**Presentation:** 3
**Contribution:** 2
**Rating:** 3
**Confidence:** 4

**Summary:**

This paper propose a new way to compress deep learning model by merging similar parameter groups within a model. The paper mainly focus on the MLP layers of the transformer model. A learned permutation is applied on to the MLP layers to be merged to minimize the difference in the merged layer output. An evaluation across all possible merging configuration is conducted to decide which layers to merge to reach the best evaluation score.

**Strengths:**

This paper explores layer merging, which is an interesting idea. The proposed method of permutation merging provides more capacity to the model after merging.

The experiments are conducted on both vision transformer models and language models

Detailed results are provided on merging different amount of layers and different layer locations.

**Weaknesses:**

Novelty-wise, weight sharing across layer is not a new concept. Early efficient language model design has explored to share weights across different transformer blocks [1], with later attemps conducted in ViTs and LLMs.

Even as a new model compression method, the proposed method seems to be not very effective, especially comparing to pruning. For example, structural pruning can achieve 2.57x lossless parameter reduction on ViT model [2], yet the proposed method can only remove 21%. Furthermore, comparing to pruning and quantization, the proposed method only reduces the amount of parameters, yet achieves no inference speedup.

One key method proposed in this work is the permute merge. Yet from the results in Figure 2 and 3, permute does not lead to significant performance improvement over the naive merging in most cases, and behave even worse on GPT-2. This leaves doubt on the effectiveness and correctness of the proposed merging technique.

The proposed method is limited to the MLP layers in the transformer model, which limits the compression ratio the model can achieve.

[1] Lan, Z. (2019). Albert: A lite bert for self-supervised learning of language representations. arXiv preprint arXiv:1909.11942.

[2] Yang, H., Yin, H., Shen, M., Molchanov, P., Li, H., & Kautz, J. (2023). Global vision transformer pruning with hessian-aware saliency. In Proceedings of the IEEE/CVF conference on computer vision and pattern recognition (pp. 18547-18557).

**Questions:**

Why permute FF merge behave worse than vanilla merge in the GPT-2 model?

Can the proposed method be extended to all model layers?

---

> ### Author Response · Authors · 2024-11-26
> **Response to Reviewer SKiK**
>
> Thanks for your time and thoughtful review of our paper, and thanks for noting our contributions of a novel method, diversity in models tested, and detailed ablations. We address the stated weaknesses and questions below:
>
> **Weakness: Weight sharing novelty**:
> The cited Albert paper is a great example of effective weight sharing for model size reduction. We emphasize that the novelty of our approach is the *post-training* integration of weight sharing. Most weight sharing approaches in the literature occur at the initialization of the model. Our work provides a realistic way to introduce weight sharing into models that have already been pre-trained, going beyond models that have to do this from the start, like Albert.
>
> **Weakness: Comparing to pruning**:
>
> Thanks for your comment regarding comparisons. We agree that even if our method takes a very different approach to compression, it should be compared to alternatives. Since we propose a general compression method not specific to ViT, we choose a general pruning method for comparison. Many recent papers on structured pruning of Transformers (which results in compression unlike many unstructured pruning methods) have centered around choosing layers for dropping and then fine-tuning the resulting model [1,2]. We use a strong baseline of 1) picking the best layers to drop after evaluation (which generalizes and strengthens many of these layer-dropping papers) and 2) fine-tuning the same as our method, for all parameters. We choose the number of layers dropped to cover a similar range as ⅓ and ½ FFs removed. We present the results in the revised PDF (Section 4.4 details, Section 5.1 results) and link them here: https://anonymous.4open.science/r/temp-C34A/README.md . We outperform or match the baseline consistently judging by the curves across different parameter reduction ratios. Although we cannot achieve exact comparisons across specific parameter reduction ratios due to the block wise reduction nature of both methods, our method trends better in general, as seen in the figure.
>
> [1] Men, et al. "Shortgpt: Layers in large language models are more redundant than you expect." arXiv preprint arXiv:2403.03853 (2024).
>
> [2] Gromov, et al. "The unreasonable ineffectiveness of the deeper layers." arXiv preprint arXiv:2403.17887 (2024).
>
> **Weakness: Permute merge gains**:
>
> The gains attributable to permutation are a smaller addition onto solely weight sharing between feed-forward sublayers. However, they do provide *consistent improvements over vanilla averaging*.  Another major contribution, as highlighted above, is the introduction of weight sharing as a *post-training* compression technique.
>
> **Weakness: MLPs only**
>
> We intentionally focus on MLP layers in this work. Our motivation is detailed in section 3.1, and we reiterate that these subcomponents are a majority of parameters for enc-only or dec-only models.
>
> **Question 1: GPT-2 permute v vanilla**:
>
> Permute FF merge does not perform worse than vanilla merge in GPT-2 (ref Figure 2, image B). However, we had a typo in from arranging results in the corresponding table in Appendix A that may have caused this confusion. We had also caught this typo on our end soon after submission and addressed it in the updated PDF. The figure in 5.1 was originally correct, and the corresponding table in the appendix is addressed to reflect this.
>
> **Question 2: all model layers**:
>
> The method was extended to all model layers in the main results, in 5.1. N-1 FFs removed is N FFs merged,which is all layers. However, we focus on smaller ranges of model layers as the method degrades at these very heavy compression ratios.

---

> > ### Comment · Reviewer_SKiK · 2024-11-27
> >
> > I would like to thank the author for the responses. I agree that I misread the permute vs vanilla result.
> >
> > However, I'm still not convinced by the effectiveness of the proposed method. First of all, the author mentions two layer-dropping papers, yet neither papers were accepted into any conferences. A quick search indicates that both papers appear to be in the "rejecting" range in the ICLR open review currently. This does not support author's point that these methods are "strong baselines". It is also not suprising that the proposed method outperforms layer dropping with the same layer selection criteria and finetuning approach: layer dropping will reduce model computation and therefore latency, but layer merging will not as the layers are still perserved. Layer merging naturally leads to a stronger model than layer dropping (with is similar to an Albert vs. only the first block of Albert).
> >
> > My main concern on the effectiveness, which is concurred by the other reviewers, is the effectiveness of the proposed layer merging vs. truly "general" model compression methods, like structural pruning. Structural pruning criteria, like Hessian, has the flexibility to explore across multiple granularities, such as filter-wise, layer-wise, or even block-wise. This leads to the effective exploration of efficiency-performance tradeoff. The proposed method, however, relies on brute-force sliding-window exploration, which cannot effectively scale up to the expontially-growing design space with a finer granularity, leading to poor tradeoff between efficiency and performance.
> >
> > I believe this is a fundamental flaw in the design of the proposed method, which cannot be bridged with revisions in this short rebuttal period. I would suggest the author to rethink about the effectiveness of layer/parameter merging, and propose more general method that can effectively merge parameters both layer-wise and in finer granularities.

---

> > > ### Author Response · Authors · 2024-12-01
> > > **Response to reviewer SKiK**
> > >
> > > **Validity of baselines**: While we don't believe that using these papers' current ICLR review scores is an appropriate way to discredit their inclusion in this work as baselines, we can just try to tell you that these two papers (and more that they have inspired) have many citations despite their recency, and are of interest to multiple communities looking at structural pruning of Transformer models, especially LLMs.
> > >
> > > **Effectiveness of method**: I have posted baseline comparisons showing superior performance of our method, and while I agree that Hessian-based pruning is a good example of more traditional pruning techniques, we can also can consider lower-cost compression techniques that straightforwardly exploit the redundant nature of large Transformer models. While it is a great feature of Hessian-based pruning that it may apply at different granularities, this not the case of all pruning methods in the literature, as many methods can apply to more specific regions (examples [1,2,3]). We also have an efficiency-performance trade-off in the number of components to merge, similar to many other compression papers where this space can be explored (and we do so in our paper). Finally, in our paper, we also show that while the sliding window technique does help choose layers to merge, random alternatives perform very well too, which can help generalize this technique further.
> > >
> > > [1] Ashkboos et al., 2024. SliceGPT: Compress Large Language Models by Deleting Rows and Columns. ICLR 2024.
> > >
> > > [2] Voita et al., 2019. Analyzing Multi-Head Self-Attention: Specialized Heads Do the Heavy Lifting, the Rest Can Be Pruned. ACL 2019.
> > >
> > > [3] Lu et al., 2024. Not All Experts are Equal: Efficient Expert Pruning and Skipping for Mixture-of-Experts Large Language Models. ACL 2024.
> > >
> > >
> > > **fundamental-flaw clarification**: The effectiveness of this method has been shown in experiments in the original draft as well as updated experiments versus a baseline in the new draft. For feedback's sake, we would appreciate if you could clarify if the fundamental flaw you are referring to means 1) the effectiveness of this method (i.e. performance), 2) specificity of the method (i.e. not as general as Hessian-based pruning), or 3) just the sliding-window approach.

---

### Official Review · Reviewer_3M7F · 2024-11-03

**Soundness:** 2
**Presentation:** 2
**Contribution:** 1
**Rating:** 3
**Confidence:** 5

**Summary:**

This work proposes a parameter merging technique that reduces the param count of feedforward layers with some fine-tuning for recovery. It consists of the permutation finding step, applying the transformations, merging transformed parameters, and finally tying the merged parameters. As an analysis, they compute similarity measures between feed-forward sublayers within the same model and find regions with highly similar activations.

**Strengths:**

- The paper does a good job delineating relevant context for neuron alignment and describing their approach.
- Also, the summary of the comparison between compression methods helps understand the trade-off of the merging method.
- Thorough analysis via ablation studies and visualization.

**Weaknesses:**

The biggest weakness is the experimental results. It seems like authors do a great job at the ablation studies and visualization, but these are secondary contributions given that this is a paper on compression method for Transformer acceleration, not interpretability research. This means the results section should cover a wider range of benchmarks and also comparisons to pruning approaches (which achieves the same end effect as merging). For example, Wanda [1] prunes 50% at one-shot (without fine-tuning) without major accuracy loss. Authors should clarify how merging is potentially more beneficial than modern pruning techniques and provide thorough comparisons & discussions.

It is nice to have experiments covering various tasks (image classification, language modeling, translation). But, I strongly encourage adding MMLU on top of simple perplexity as it better demonstrates language modeling. Also, LLaMA models should be tested given their overwhelming popularity over GPT-2 -- it will be a bigger contribution to the community.

[1] M. Sun, et al., "A simple and effective pruning approach for large language models", ICLR 2024.

**Questions:**

1. Typo in line 89: "These same patterns do not in counterpart attention sublayers." ?
2. As described above, authors should compare their merging method to pruning and also quantization approaches. I see that it explores how merging can be combined with quantization (Table 4), but quantization should be compared head-to-head with their merging method, as both are used to reduce storage.

---

> ### Author Response · Authors · 2024-11-26
> **Response to Reviewer 3M7F**
>
> Thanks for your time and thoughtful review of our paper, and thanks for noting our contributions of ablations, visualizations, and contextualizing our method with prior work and other compression methods. We address the stated weaknesses and questions below:
>
> **Weakness 1: pruning comparisons**:
>
> Thanks for your suggestion regarding comparisons. We have included a new layer-pruning baseline that realizes compression ratios similar to ours, and discussed the choice of this baseline in new subsection 4.4 in our updated PDF. We do not consider methods like Wanda as baselines due to their unstructured sparsity patterns. Although it can achieve 50% sparsity in some LLMs, model weights are not actually compressed with this sparsity pattern without specific storage considerations (like COO, CSR, DOK sparse formats via sparse libraries), and these generally require ratios > 50% sparsity for actual compression on 2D weight matrices. This discussion of merging, unstructured pruning, and structured pruning is expanded in the PDF.
>
> In our revision, we use a strong layer-pruning baseline of 1) picking the best layers to drop after evaluation which generalizes and strengthens many proposed layer-pruning methods [1,2] and 2) fine-tuning the same as our method, for all parameters. We choose the number of layers dropped to cover a similar range as ⅓ and ½ FFs removed. We present the results in the revised PDF (Section 4.4 details, Section 5.1 results) and link them here: https://anonymous.4open.science/r/temp-C34A/README.md. We outperform or match the baseline consistently judging by the curves across different parameter reduction ratios. Although we cannot achieve exact comparisons across specific parameter reduction ratios due to the block wise reduction nature of both methods,  our method trends better in general, as seen in the figure.
>
> [1] Men, et al. "Shortgpt: Layers in large language models are more redundant than you expect." arXiv preprint arXiv:2403.03853 (2024).
>
> [2] Gromov, et al. "The unreasonable ineffectiveness of the deeper layers." arXiv preprint arXiv:2403.17887 (2024).
>
> **Weakness 2: Benchmarking GPT-2**:
>
> Thanks for your suggestion regarding LM type and evals. While we agree that LLaMA models would be ideal due to their popularity, they are still quite large for our experimentation, whereas GPT-2 Large is much smaller and still performant for its size. Regarding zero-shot eval, we wholeheartedly agree that for large language models, evals like MMLU are important to test knowledge retention alongside language modeling performance. However, on a smaller model like GPT-2, evaluating on MMLU is more appropriate after fine-tuning (like in the original MMLU paper). Since we are dealing with smaller scale models, and trying to measure just language modeling capability rather than also world knowledge, which is the goal of MMLU, we believe PPL is a sufficient metric in our reduced parameter case.
>
> **Question 2: Comparisons to pruning and quantization**
> We have now added a strong pruning baseline, and respectfully disagree with comparing directly with quantization. Merging and quantization address orthogonal dimensions of compression (precision, redundancy), and we demonstrate their complementary and orthogonal performance, yielding benefits greater than either method alone.
>
> Finally, thank you for your careful reading and catching our typo, it is addressed in our revision.

---

### Official Review · Reviewer_zq1y · 2024-11-08

**Soundness:** 3
**Presentation:** 4
**Contribution:** 3
**Rating:** 5
**Confidence:** 4

**Summary:**

The paper proposed a novel model compression approach by merging parameters. Specifically, the parameters from some of the linear layers are averaged after a set of permutations, which involves to maximize overall correlation of the inputs of layers. This method aims to reduce the parameter storage costs of deep neural networks, particularly Transformers, by merging similar parameter groups within the model rather than pruning individual parameters.

**Strengths:**

The paper is well-written with a clear and logical structure. The paper presents a novel method to reduce the storage costs of deep Transformer-based models by merging their parameters. The experimental results provide a detailed discussion of parameter merging across multiple deep models on various tasks, demonstrating the effectiveness of the proposed approach.

**Weaknesses:**

As shown in Table 1, parameter merging maintains the model's inference speed but still requires fine-tuning, highlighting the drawbacks of this approach. Despite the distinct from parameter pruning, parameter merging/sharing remains a common model compression technique. However, the paper lacks of experimental comparison and discussion with other parameter pruning methods, such as [1], weakens the argument presented in this paper. Notably, [1] achieves a nearly unchanged ViT accuracy (-0.07, 83.36 → 83.29) while reducing model parameters by over 40 percent, including a 1.9x run time speedup. In contrast, the paper reports a significant ViT precision drop (-1.1, 80.3 → 79.2) with a parameter reduction of about 20 percent and no improvement in inference speed.
[1]: Global Vision Transformer Pruning with Hessian-Aware Saliency, Huanrui Yang et. al, CVPR 2023.

**Questions:**

1. Have there been any attempts to directly compare the proposed method with a parameter-sharing structure trained from scratch? Essentially, the "compressed model" relies on a shared parameter structure following fine-tuning. Given that fine-tuning is an integral part of the proposed approach, could a "compression from scratch" strategy potentially yield better results?
2. Could the authors clarify the reasons behind adopting the sliding window strategy for selecting sub-layers? Are there potentially better designs for this selection process? While the sliding window approach appears straightforward, it may lack novelty and clear motivation.

---

> ### Author Response · Authors · 2024-11-26
> **Response to reviewer zq1y**
>
> Thanks for your time and thoughtful review of our paper, and thanks for noting our contribution of our novel compression method and its experimental effectiveness. We address the stated weaknesses and questions below:
>
> **Weakness**: Thanks for your comment regarding comparisons and discussions. We agree that even if our method takes a very different approach to compression, it should be compared to alternatives. Since we propose a general compression method not specific to ViT, we choose a general pruning method for comparison. Many recent papers on structured pruning of Transformers (which results in compression unlike many unstructured pruning methods) have centered around choosing layers for dropping and then fine-tuning the resulting model [1,2]. We use a strong baseline of 1) picking the best layers to drop after evaluation (which generalizes and strengthens many of these layer-dropping papers) and 2) fine-tuning the same as our method, for all parameters. We choose the number of layers dropped to cover a similar range as ⅓ and ½ FFs removed. We present the results in the revised PDF (Section 4.4 details, Section 5.1 results) and link them here for quick viewing: https://anonymous.4open.science/r/temp-C34A/README.md . We outperform or match the baseline consistently judging by the curves across different parameter reduction ratios. Although we cannot achieve exact comparisons across specific parameter reduction ratios due to the block wise reduction nature of both methods, our method trends better in general, as seen in the figure.
>
> [1] Men, et al. "Shortgpt: Layers in large language models are more redundant than you expect." arXiv preprint arXiv:2403.03853 (2024).
>
> [2] Gromov, et al. "The unreasonable ineffectiveness of the deeper layers." arXiv preprint arXiv:2403.17887 (2024).
>
> **Question 1: Training from scratch**:
> Parameter sharing from scratch is a useful framework for pre-encoding parameter efficiency that has seen success in the past, as discussed in Section 2.1. However, our method showcases the effectiveness of weight sharing as a *lightweight, post-training compression method* that applies to pre-trained models, whereas the methods discussed in Section 2.1 train models from scratch. To retrain the models in this work with this new sharing structure would *require extensive pretraining* which is out of scope of this work.
>
> **Question 2: Adjacency**:
> We use the sliding window strategy to exhaust all sets of k adjacent feed-forward layers. We choose adjacency for 1) combinatorial ease (ex, 36 GPT-2 layers choose 12 is > 1B) as well as 2) evidence of similarity aligning with adjacency from prior work [1, 2] as well as our own (i.e. Figure 4).
>
> [1] Pires et al. "One wide feedforward is all you need." arXiv preprint arXiv:2309.01826 (2023).
>
> [2] Kornblith, Simon, et al. "Similarity of neural network representations revisited." International conference on machine learning (ICML), 2019.

---

### Author Response · Authors · 2024-12-02
**Follow-up**

Dear Reviewers,

Thank you for your time and effort in reviewing our paper. As today is the final day for responding to authors, we would appreciate it if you could evaluate our responses to your comments and concerns, as well as our updated paper with new baselines and discussion of pruning methods versus merging methods (all in red in the PDF). If you feel that we have addressed your concerns to some degree, we encourage you to update your score accordingly. Additionally, we value your feedback as it helps us improve our work, so should you have additional questions or comments, we will try our best to respond to them with the remaining time left.

Best regards

---

### Note · Authors · 2024-12-13

I have read and agree with the venue's withdrawal policy on behalf of myself and my co-authors.